# Structure and Vibrational Spectra of Pyridine Solvated Solid Bis(Pyridinesilver(I) Perchlorate, [Agpy₂ClO₄]·0.5py

**Nóra V. May [1], Niloofar Bayat [1,2], Kende Attila Béres [1,3], Petra Bombicz [1], Vladimir M. Petruševski [4], György Lendvay [1], Attila Farkas [5] and László Kótai [1,6,*]**

[1] Research Centre for Natural Sciences, Hungarian Academy of Sciences, Magyar Tudósok krt. 2., 1117 Budapest, Hungary
[2] Department of Inorganic and Analytical Chemistry, Budapest University of Technology and Economics, Műegyetem rkp. 3., 1111 Budapest, Hungary
[3] Institute of Chemistry, ELTE Eötvös Loránd University, Pázmány Péter s. 1/A, 1117 Budapest, Hungary
[4] Faculty of Natural Sciences and Mathematics, Ss. Cyril and Methodius University, 1000 Skopje, North Macedonia
[5] Department of Organic Chemistry and Technology, Budapest University of Technology and Economics, Műegyetem rkp. 3., 1111 Budapest, Hungary
[6] Deuton-X Ltd., Selmeci u. 89, 2030 Érd, Hungary
[*] Correspondence: kotai.laszlo@ttk.hu

**Simple Summary:** It is vitally important that scientists are able to describe their work simply and concisely to the public, especially in an open-access on-line journal. The simple summary consists of no more than 200 words in one paragraph and contains a clear statement of the problem addressed, the aims and objectives, pertinent results, conclusions from the study and how they will be valuable to society. This should be written for a lay audience, i.e., no technical terms without explanations. No references are cited and no abbreviations. Submissions without a simple summary will be returned directly. Example could be found at http://www.mdpi.com/2076-2615/6/6/40/htm.

**Abstract:** A hemipyridine solvate of bis(pyridine)silver(I) perchlorate, [Agpy₂ClO₄]·0.5py (compound **1**) was prepared and characterized by single crystal X-ray analysis and vibrational spectroscopy (R and low-temperature Raman). Compound **1** was prepared via the trituration of [Agpy₂ClO₄] and 4[Agpy₂ClO₄]·[Agpy₄]ClO₄ (as the source of the solvate pyridine) in a mixed solvent of acetone:benzene =1:1 (v = v) at room temperature. The monoclinic crystals of compound **1** were found to be isomorphic with the analogous permanganate complex (a = 19.1093(16) Å, b = 7.7016(8) Å, c = 20.6915(19) Å, β = 105.515(7)°; space group: C2/c). Two [Agpy₂]⁺ cations formed a dimeric unit [Agpy₂ClO₄]₂, and each silver ion was connected to two ClO₄⁻ anions via oxygen atoms. The Ag···Ag distance was 3.3873(5) Å, the perchlorate ions were coordinated to silver ions, and the Ag···O distances were 2.840(2) Å and 2.8749(16) Å in the centrosymmetric rectangle of Ag-O-Ag-O. The stoichiometric ratio of the monomer [Agpy₂ClO₄] and the solvent pyridine was 1:0.5. The guest pyridine occupied 527.2 Å³, which was 18.0% of the volume of the unit cell. There was no additional residual solvent-accessible void in the crystal lattice. The solvate pyridine was connected via its a-CH to one of the O atoms of the perchlorate anion. Correlation analysis, as well as IR and low-temperature Raman studies, were performed to assign all perchlorate and pyridine vibrational modes. The solvate and coordinated pyridine bands in the IR and Raman spectra were not distinguishable. A perchlorate contribution via Ag-O coordination to low-frequency Raman bands was also assigned.

**Keywords:** pyridine complexes; silver perchlorate; vibrational spectroscopy; single crystal structure; hydrogen bond

## 1. Introduction

The ligand–anion reactions of transition metal tetraoxometallates containing reducing ligands (ammonia, pyridine or urea) [1–5], including silver complexes containing ammonia and pyridine [6–11], are intensively studied areas of coordination chemistry. The interest in these compounds is generated by the structural diversity of [Agpy$_n$]XO$_4$ complexes (*n* = 2, 2.4, 2.5, and 4, X = Cl, Mn, and Re, respectively), including variability in the silver coordination number, solvate formation, mixed crystal formation and the presence/absence of Ag-Ag argentophilic and pyridine $\alpha$-C-H···O-XO$_3$ hydrogen bond formation [12,13]. The effects of these interactions on the coordination modes, topology and crystal engineering characteristics of Ag$^I$ complexes have enormous importance [14–20].

Among the 12 possible members of the four series of the [Agpy$_n$]XO$_4$ compounds (*n* = 2, 2.4, 2.5 and 4, X = Cl, Mn, and Re, respectively), 10 compounds have been isolated until now [9–13]. The existence of [Agpy$_{2.5}$]XO$_4$ compounds with 0.5 pyridine solvate content (X = Cl (compound **1**) and Mn (compound **2**)) were recently discovered [12,13]. Among the four existing [Agpy$_n$]ClO$_4$ complexes, three have already been structurally characterized (*n* = 2 [21], 2.4 [22], and 4 [23] for compounds **3**, **4** and **5**, respectively). Earlier attempts to synthesize the missing member of the series, with *n* = 2.5, were not successful. In this paper, filling the gap, we report on the synthesis and structural characterization of [Agpy$_2$ClO$_4$]·0.5py) (compound **1**) in order to explore the possible enhanced structural variability of this solvate compared with the permanganate analog. Compound **1** was prepared at a satisfactory purity to perform vibrational spectroscopic characterization including correlation analysis.

## 2. Results and Discussion

### 2.1. Preparation of [Agpy$_2$ClO$_4$]·0.5py (Compound **1**)

The reaction of silver nitrate with aq. pyridine solutions generally resulted in a mixture of compounds **3** and **4**. When we used 6 equivalents of pyridine to silver at a 36.5% pyridine concentration by cooling the reaction mixture to 10 °C, a mixture of compound **1**, compound **3** and compound **4** was formed (1:3:15 ratio), but all of our efforts to isolate pure compound **1** in this reaction route failed [13]. The 1:1 mixture of compounds **3** and **4**, prepared in a similar way applying an Ag:py ratio of 1:3 and a pyridine concentration of 1.25% at 10 °C, however, proved to be a good starting material in the preparation of compound **1** with trituration in a 1:1 (*v/v*) acetone–benzene mixture at room temperature [13], during which colorless block-shaped crystals suitable for single-crystal X-ray study were grown in 2 days. The solid phase left back was compound **1**, whereas the solid phase isolated by the evaporation of the mother liquor was a mixture of compounds **1** and **3**. The solid compound **1** that formed was only stable in contact with the acetone–benzene mother liquor. Its powder XRD results (Figure S1) agreed well with the calculated one from the single crystal measurement.

### 2.2. Structure of [Agpy$_2$ClO$_4$]·0.5py (Compound **1**)

The monoclinic crystals of compound **1** were found to be isomorphic with the analogous permanganate complex (compound **2**) [13]. Their space group was *C2/c*, *a* = 19.1093(16) Å, *b* = 7.7016(8) Å, *c* = 20.6915(19) Å, *β* = 105.515(7)°, *V* = 2934.2(5) Å$^3$, *T* = 103(2) K, *Z* = 4, and *D$_x$* = 1.834 Mg/m$^3$) (Table S1). These values were very close to those found for compound **2** (Table 1). The cell volume of compound **1** was 5.5% smaller than the cell volume of compound **2**, as was expected. An opposite tendency was observed in the case of isomorphic [4Agpy$_2$XO$_4$.Agpy$_4$XO$_4$] compounds (X = Cl and Mn) [10,13]. This means that not only the size of anion but also other conformational effects have influence on the compactness of the crystal lattices of [Agpy$_n$]XO$_4$ (X = Cl, Mn) compounds.

**Table 1.** Comparison of main cell parameters and inter- and intramolecular distances of [Ag-py$_2$ClO$_4$]·0.5py (compound **1**) and [Agpy$_2$MnO$_4$]·0.5py (compound **2**).

|  | Compound 1 | Compound 2 |
|---|---|---|
| Temperature | 103(2) K | 293(2) K |
| Crystal system | monoclinic | monoclinic |
| Space group | *C*2/c | *C*2/c |
| Unit cell dimensions |  |  |
| *a* | 19.1093(16) Å | 19.410(1) Å |
| *b* | 7.7016(8) Å | 7.788(1) Å |
| *c* | 20.6915(19) Å | 21.177(1) Å |
| β | 105.515(7)° | 104.20(1)° |
| Cell volume | 2934.2(5) Å$^3$ | 3103.4(1) Å$^3$ |
| Anion | ClO$_4^-$ not disordered | MnO$_4^-$ disordered 66:34 |
| KPI | 69.8% | 67.4% |
| Pyridine | 527.2 Å$^3$, 18.0% | 604.5 Å$^3$, 19.5% |
| Ag···Ag | 3.3873(5) Å | 3.4213(9) Å |
| Ag-O2 | 2.840(2) Å | 2.548(14) Å |
| Ag-O2′ | 2.8749(16) Å | 2.745(9) Å |
| C$\alpha$21-H21$\alpha$···O2$_{XO4}$ | 2.355 Å | 2.602 Å |
|  | 2.701 Å | 2.770 Å |
| π−π N1···N1 | 3.745 Å | 3.782 Å |
| π−π N1···N2 | 3.666 Å | 3.740 Å |

The asymmetric unit was found to contain the [Agpy$_2$ClO$_4$] moiety, as well as half a pyridine, arranged by a twofold axis. The $Z'_{pyridine}$ = 0.5 (Z = 4) symmetry codes to generate equivalent atoms C31i and C32i were −x + 1, y, −z + 1/2 + 1 (Figure 1). Two pyridine molecules were shown to coordinate to the Ag$^+$ cation. The Ag-N distances were 2.159(2) Å and 2.161(2) Å, only a bit shorter than in the permanganate complex (2.166(esd) Å and 2.174(esd) Å, respectively).

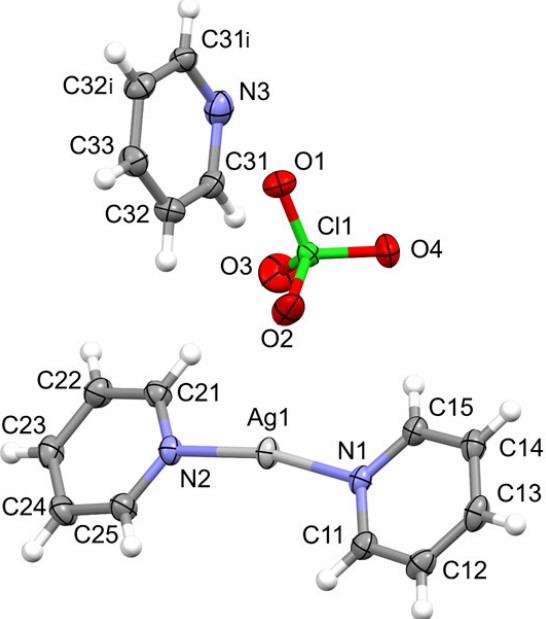

**Figure 1.** ORTEP presentation of the molecular structure and atomic numbering scheme of compound **1** (symmetry codes to generate equivalent atoms C31i and C32i were −x + 1 and y, −z + 1/2 + 1, respectively). The displacement ellipsoids are drawn at the 50% probability level.

Two [Agpy₂]⁺ cations were found to form a dimeric unit [Agpy₂ClO₄]₂ (Figure 2a), where silver ions were connected by the O2 oxygen atoms of two ClO₄⁻ anions. The Ag⋯Ag distance was 3.3873(5) Å, whereas the Ag⋯O distances were 2.840(2) Å and 2.8749(16) Å in the centrosymmetric rectangle of Ag-O-Ag-O (Figure 2b and Table 1).

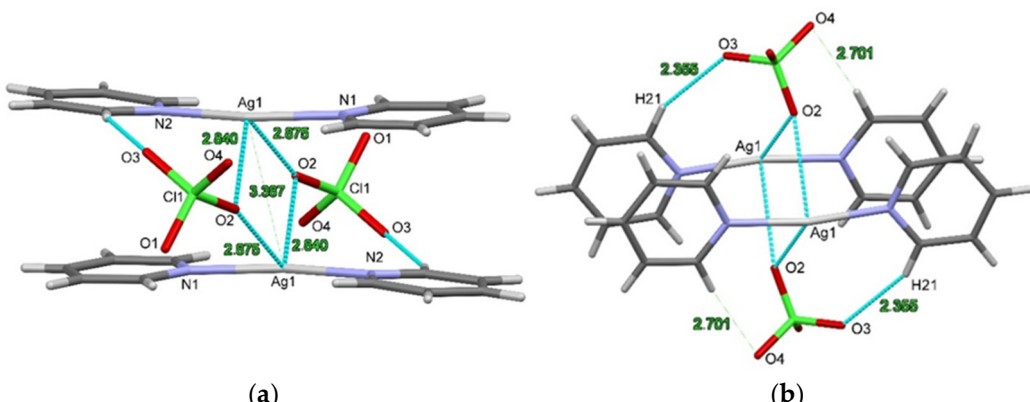

(**a**)                                                (**b**)

**Figure 2.** The centrosymmetric dimer of [Agpy₂ClO₄]₂ (**a**) and the Cα-H(pyridine ligand) ⋯O(perchlorate) interactions (**b**) in the dimer of compound **1**.

The Cα-H(pyridine ligand)⋯O(perchlorate) intramolecular hydrogen bonds were shown to support complex formation (Table 2). The H⋯O distance in C21-H21⋯O3 was 2.355 Å. On the other side of the silver cation, the C15-H15⋯O4 distance was long and the H⋯O distance was 2.701 Å (2.770 Å in compound **2**), as O4 could not be close to the pyridine due to steric reasons (Figure 2b).

**Table 2.** Hydrogen bond interactions (distances and angles are given) in compound **1**.

| D-H⋯A | D-H [Å] | H⋯A [Å] | D⋯A [Å] | D-H⋯A [°] | Symmetry Operation |
|---|---|---|---|---|---|
| C21-H21⋯O3 | 0.93 | 2.360 | 3.228(3) | 156 | intra |
| C15-H15⋯O4 | 0.93 | 2.701 | 3.621(3) | 170 | intra |
| C24-H24⋯O4 | 0.93 | 2.598 | 3.261(3) | 129 | x, 1−y, −1/2 + z |
| C13-H13⋯O1 | 0.93 | 2.600 | 3.376(3) | 141 | −1/2 + x, 1/2 + y, z |
| C23-H23⋯O1 | 0.93 | 2.595 | 3.504(3) | 166 | x, 1−y, −1/2 + z |
| C32py⋯H32⋯O4 | 0.93 | 2.64 | 3.555(3) | 169 | x, 1 + y, z |

The atomic coordinates, anisotropic displacement parameters, molecular distances/angles, and hydrogen bond distances are given in Tables S1–S6.

The conformation of the [Agpy₂]⁺ cations in compounds **1** and **2** were found to be similar (rmsD = 0.1008 and maxD = 0.1572 Å) [24]. However, the geometry of the whole complex molecule was considerably different due to the effect of the exchange of the anion from MnO₄⁻ to ClO₄⁻ (Figure 3).

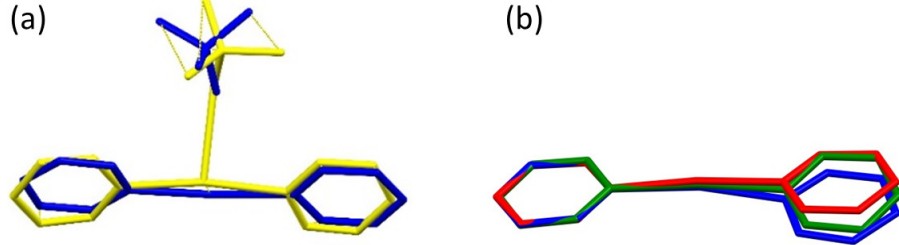

(a)                                                (b)

**Figure 3.** Comparison of (**a**) [Agpy₂]ClO₄·0.5py (compound **1**) in blue and [Agpy₂]MnO₄·0.5py (compound **2**) in yellow; (**b**) [Agpy₂]ClO₄·0.5py (compound **1**) in blue, [Agpy₂]ClO₄ (compound **3**) in red, and [Agpy₄]ClO₄·4[Agpy₂]ClO₄ (compound **4**) in green.

The structural dissimilarities induced conformational variability of the [Agpy$_2$]$^+$ cation with the same anion (perchlorate) in the crystal structures of [Agpy$_2$ClO$_4$]·0.5py (compound **1**), [Agpy$_2$ClO$_4$] (compound **3**), and [Agpy$_4$]ClO$_4$·4[Agpy$_2$ClO$_4$] (compound **4**) (Figure 3). The rmsD values were found to be 0.1334 and 0.1248 and the maxD values were found to be 0.2344 and 0.2019 when the conformations of [Agpy$_2$]$^+$ cations in compound **1** were compared to those in compound **3** [21] or compound **4** [22].

The Ag$^+$ cation was shown to have trigonal-bipyramid coordination in compound **1** when we also considered the oxygen atoms at a distance of around 3 Å. Two pyridine nitrogen atoms were found in the axial positions, while O2, O4, and a symmetry-generated O2 were shown to occupy the equatorial plane (Figure 4a). Via the longest Ag-O4 coordination bond, the dimers formed a chain along the c crystallographic axis (Figure 4b). Each oxygen atom of the perchlorate anion was shown to have a distinctive role in the formation of the crystal lattice. O1 forms hydrogen bonds with the pyridine para-hydrogens (C13-H and C23-H; see Figure 4c), which construct a chain of dimers in the ac diagonal. O2 takes part in the formation of the dimer (Figure 2). O3 is in a hydrogen bond with C$\alpha$ of the ligand pyridine, stabilizing the dimer (Figure 4c). O4 is part of the 1D chain of the coordination network; it is in a weak intradimer hydrogen bond with a ligand pyridine C$\alpha$-H15, and it was shown to be close to a neighboring pyridine meta C24-H24. Finally, O4 was found to form a weak hydrogen bond C32-H32···O4 to the solvent pyridine (Table 2). The position of the complex dimers in their 1D chain was alternating, with an angle of 88.60° concerning the rings containing N1 and N2 (Figure 4c).

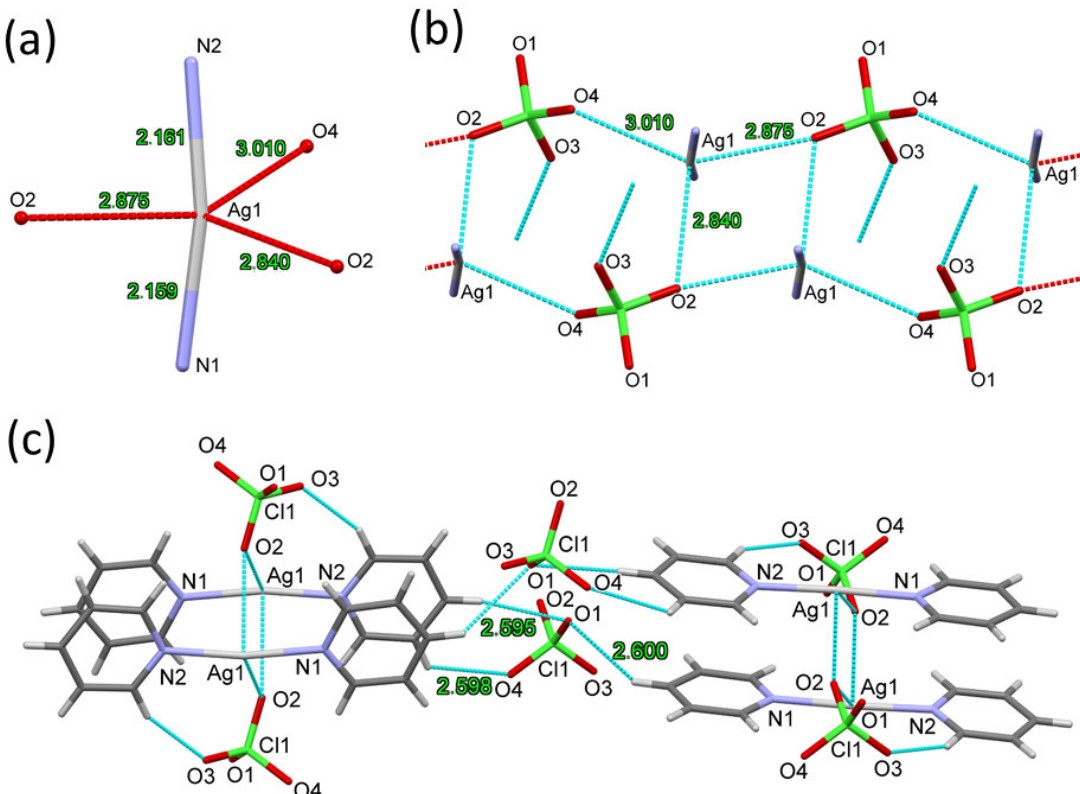

**Figure 4.** The coordination sphere of the Ag$^+$ cation in compound **1** (**a**) and the 1D coordination network of compound **1** (**b**) with the chain of dimers in the ac diagonal formed by O1 and O4 (**c**).

The neighboring dimers were found to be connected by the weak C13-H13···O1, C23-H23···O1 and C24-H24···O4 intermolecular interactions (a), and the almost perpendicular alternating positions of the dimers of compound **1** were shown to be built up along the ac diagonal (b) (Figure 4c). $\pi$···$\pi$ interactions were shown to contribute to the

stability of the crystal lattice of compound **1** (Figure 5). The distance between the aromatic rings containing N1 and N2 was found to be 3.6657(14) Å within the dimer, while the distance between the aromatic rings of neighboring dimers containing N1 and N1 was calculated as 3.7449(14) Å. The angle of ring centroids in the column was 151.93°.

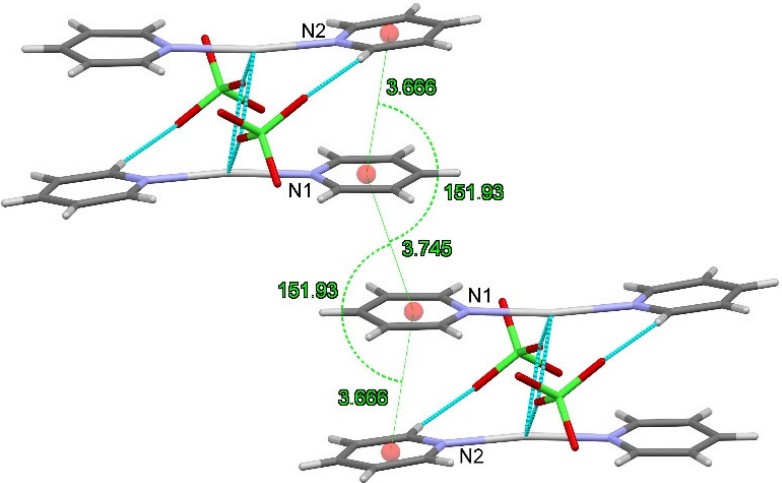

**Figure 5.** The π···π interactions formed between pyridine ligands in compound **1**. The distances of the ring centroids are indicated.

The stoichiometric ratio of the monomer [Agpy$_2$ClO$_4$] and the solvent pyridine was calculated as 1:0.5. The guest pyridine occupied 527.2 Å$^3$, which was 18.0% of the volume of the unit cell (Figure 6). There was no additional residual solvent-accessible void in the crystal lattice. The solvate pyridine was found to be connected to the O4 atom of the perchlorate anion with the $\alpha$-CH. The Kitaigorodskii packing coefficient was found to be 69.8% [25], and it dropped to 59.5% after the removal of the solvent from the crystal lattice. The packing arrangement in the crystals of compound **1** viewing from the a, b and c crystallographic axes is shown in Figure S2, which includes the solvent pyridine.

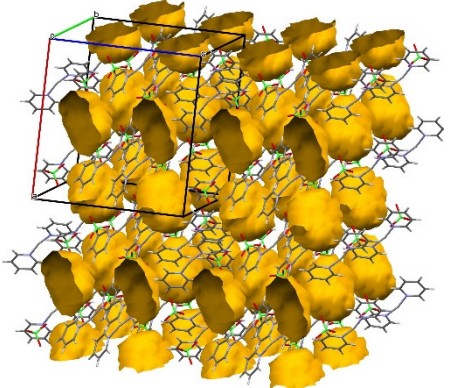

**Figure 6.** The voids accessible for the solvent pyridine in compound **1**. The voids of the solvent molecules occupy 18.0% of the unit cell.

The packing arrangement in crystals of compound **1** viewed along the *a*, *b* and *c* crystallographic axes is shown in Figure 7, which includes the solvent pyridine.

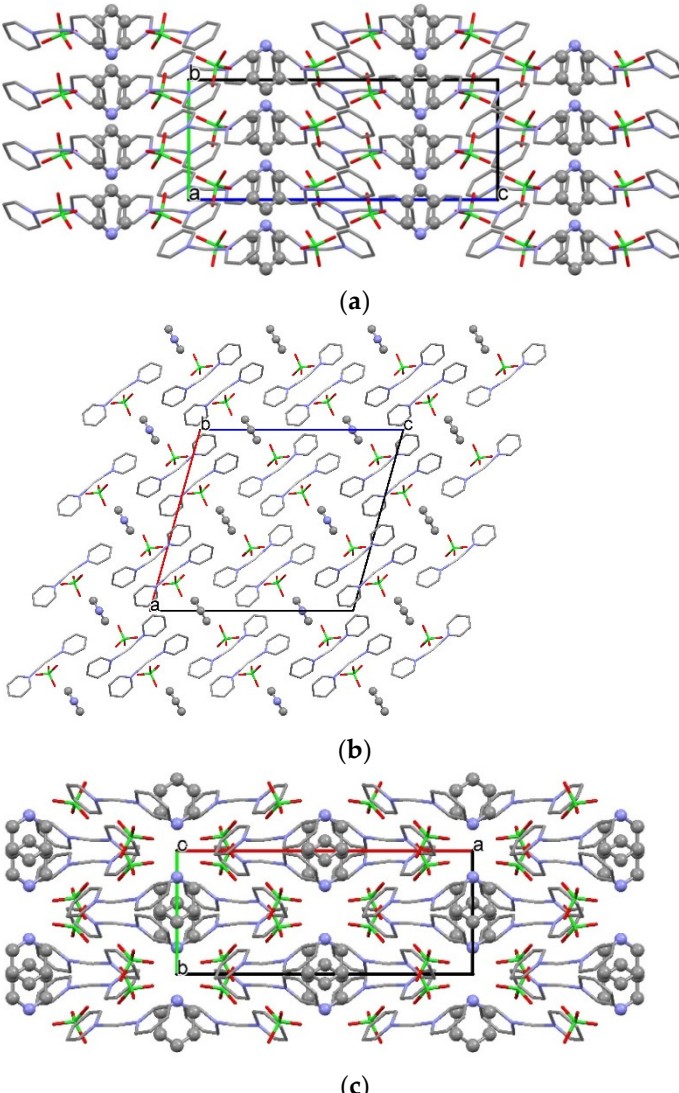

**Figure 7.** The packing arrangement of compound **1** (capped sticks) and the solvent pyridine (ball and stick representation). (**a**) View along the *a* crystallographic axis; (**b**) view along the *b* crystallographic axis; (**c**) view along the *c* crystallographic axis.

### 2.3. Vibrational Spectroscopic Analysis of [Agpy₂ClO₄]·0.5py (Compound **1**)

Based on the single-crystal X-ray study (see preceding paragraphs), compound **1** was found to have a monoclinic cell (space group *C2/c*, *Z* = 4 for the full unit cell) but with *Z* = 2 for the primitive cell. We were able to perform a correlation analysis for the perchlorate ion, the isolated silver cation (only translational modes are taken into consideration), and two kinds of pyridine, coordinated ($C_1$ site) and non-coordinated solvate pyridine ($C_2$ site). The correlation diagrams for the internal and the external vibrational modes of perchlorate ions, pyridines, and translational modes of silver ion are given in Figures 8 and S3–S7.

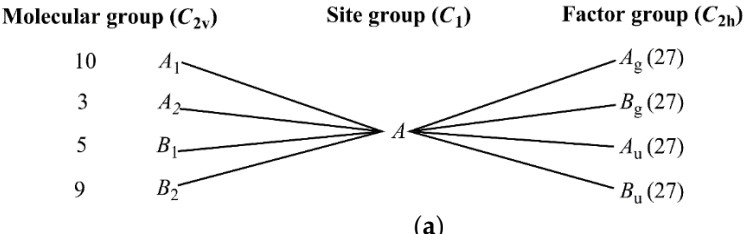

(**a**)

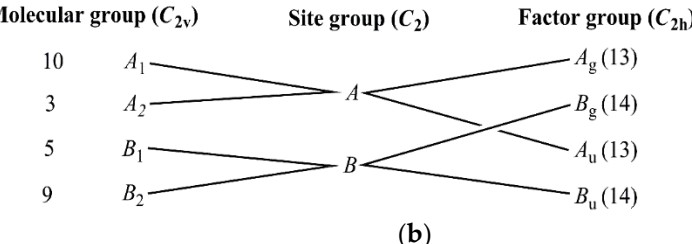

**Figure 8.** Correlation table for the 27 internal vibrational modes of the coordinated type of pyridine rings at $C_1$ (**a**) and $C_2$ (**b**) sites.

We found nine internal vibrational modes of the $ClO_4^-$ anions at $C_1$ sites. Among them, the *E* mode was doubly and the *F2* modes were triply degenerate. The $A_u$ and $B_u$ modes of the factor group were IR active, whereas the $A_g$ and $B_g$ were Raman active. Altogether, 18 IR ($9A_u$ and $9B_u$) and 18 Raman ($9A_g$ and $9B_g$) vibrational degrees of freedom could be expected. Correlation diagrams for the external vibrations of the $ClO_4^-$ anions at $C_1$ sites showed that a total of 24 such degrees of freedom (12 of translational and 12 of rotational origin) exist. The bands associated with these modes are expected in the far-IR region (12 bands, $6A_u$ and $6B_u$) and in the low-frequency part of the Raman spectra (12 bands, $6A_g$ and $6B_g$). *R* and *T* denote hindered rotations and hindered translations of the anions (ESI Figures S3–S5 and S7), respectively. No predictions of the intensities (either here or later) could a priori be made.

A total of 12 vibrational degrees of freedom were found to exist (four $Ag^+$ ions per primitive cell). All *g*-modes were active in Raman scattering, and all *u*-modes were IR active.

Each mode from the local (site) group was shown to split into four components in the factor group. There were $27A_g$, $27B_g$, $27A_u$, and $27B_u$ modes giving rise to 108 vibrational degrees of freedom. We observed two types of pyridine rings at $C_1$ sites, so the number of modes had to be doubled, i.e., $54A_g$, $54B_g$, $54A_u$ and $54B_u$. Each external mode from the local (site) group split into four components in the factor group. There were $6A_g$, $6A_u$, $6B_g$, and $6B_u$ external modes giving rise to 24 vibrational degrees of freedom (12 of translational and 12 of rotational origin). There were two types of pyridine rings at $C_1$ sites, so the number of modes had to be doubled, i.e., $12Ag$, $12B_g$, $12A_u$ and $12B_u$ modes.

The correlation tables for the 27 internal and 12 external vibrational modes of the coordinated and non-coordinated pyridine rings at the $C_1$ and $C_2$ sites are given at Figures 8a,b, S3 and S4. Among the abovementioned modes of pyridine rings (that is $99A_g$, $102B_g$, $99A_u$, and $102B_u$), $1A_u$ and $2B_u$ were found to be acoustic modes. Since these acoustic modes under the **k** = 0 approximation had a frequency of 0, the $99A_g$, $102B_g$, $98A_u$ and $100B_u$ bands, at most, were expected due to fundamental transitions that appeared in the Raman (201) and the IR (198) spectra. Bands due to second-order transitions could further complicate the spectral picture, especially in cases where conditions for Fermi resonance are fulfilled.

### 2.4. Assignment of the Vibrational Modes in the Spectra of Compound **1**

To support the assignment of the spectral bands, density functional theory (DFT) calculations were performed on the pyridine ligand, the dimer of $Agpy_2ClO_4$ shown in Figure 2 , as well as the dimer coordinated by two py ligands at the O4 atoms, referred to a "dimer and 2 py" from here on. The geometries optimized at the M05/LANL2DZ and 6-31G(d,p) level (see the Methods section) are listed in Tables S7 and S8. In Table 3, we present the vibrational frequencies obtained for the pyridine modes.

The symmetric stretching and bending modes of perchlorate ions ($v_1$ and $v_2$) were not shown to be IR active for the regular tetrahedral perchlorate ion, but these modes did appear in the IR spectrum of compound **1** because of the distortions (ESI Figure S6) (see correlation analysis above). The weak singlet band was shown to belong to the symmetric

stretching mode and appeared at 929 cm$^{-1}$ in the IR spectrum (Figure 9). Accordingly, an intensive band appeared at 930 cm$^{-1}$ in the Raman spectra of compound **1** (Figure 10). The DFT calculations on the dimer predicted a doublet at 960 and 961 cm$^{-1}$ for this mode, in which the symmetric vibrations of the two perchlorate units were coupled in-phase and anti-phase, the latter displaying some intensity. The deviation in the frequencies is within the expected range considering the capabilities of DFT methods.

**Table 3.** Assignment of the pyridine normal modes in the IR and Raman spectra of compound **1**. * Overlapped with Cl-O modes.

| $C_{2v}$ | $\nu$ | Vibrational Mode | Compounds | | | | Pyridine [26–28] | | DFT Calculations | | |
|---|---|---|---|---|---|---|---|---|---|---|---|
| | | | **1** | | **4 [8]** | | | | Pyridine | Agpy$_2$ClO$_4$ Dimer | Dimer and 2 py |
| | | | IR | Raman | IR | Raman | IR | Raman | | | |
| A$_1$ (in-plane) | 1 | C-H stretch | 3120 | 3150 | 3094 | 3082 | 3076 | 3076 | 3261 | 3268–3272 | 3258–3269 |
| | 2 | C-H stretch | 3074 | 3085 | 3062 | 3066 | 3060 | 3060 | 3236 | 3236–3242 | (3218–3243)[b] |
| | 3 | C-H stretch | 3046 | 3068 | 3043 | 3030 | 3030 | 3030 | 3191 | 3223, 3229 | (3218–3238)[b] |
| | 4 | Ring stretch | 1574 | 1586 | 1592 1571 | 1602 1571 | 1578 | 1578 | 1652 | 1668–1670 | 1660–1673 |
| | 5 | Ring stretch | 1488 | 1488 1485 | 1482 | 1483 | 1483 | 1483 | 1479 | 1515–1520 | 1519–1524 |
| | 6 | C-H wag | 1222 | 1224 | 1215 | 1215 | 1217 | 1217 | 1246 | 1247–1257 | 1244–1267 |
| | 7 | C-H wag | 1069[a] | 1062[a] | 1079 | 1072 | 1071 | 1071 | 1094 | 1088, 1096, 1097[b] | 1095, 1097 1103 |
| | 8 | Ring bend | 1040[a] | 1042[a] | 1034 | 1034 | 1031 | 1031 | 1002 | 1024–1027 | 1006, 1024, 1026, 1029, 1030 |
| | 9 | Ring breathing | 880 | 863 | - | 865 | 858 | 858 | 1046 | 1024–1027 | 1043-1049 |
| | 10 | Ring bend | 619[a] | 621[a] | - | - | 601 | 601 | 606 | 630–636 | 610, 611, 632–640 |
| A$_2$ (out of plane) | 11 | C-H wag | 983 | 992 | 983 | 985 | 982 | 982 | 1001 | 1020–1023 | (1004–1040)[c] |
| | 12 | C-H wag | 881 | - | 885 | 878 | 887 | 887 | 897 | 897–903 | 898–929 |
| | 13 | Ring bend | - | - | - | - | - | - | 375 | 389–395 | 389–405 |
| B$_1$ (out of plane) | 14 | C-H wag | 1014 | 1013 | 1003 | 1010 | 997 | 997 | 1005 | 1011–1015 | (1005–1016)[c] |
| | 15 | C-H wag | 949 | 929 * | 944 | - | - | - | 966 | 988–991 | 973–996 |
| | 16 | C-H wag | 751 | - | 751 | - | - | - | 713 | 714–717 | 715–729 |
| | 17 | Ring bend | 704, 692, 674 | - | 700 | - | - | - | 763 | 765–770 | 767–783 |
| | 18 | Ring bend | - | - | - | - | - | - | 412, | 416–420 | 415–425 |
| B$_2$ (in-plane) | 19 | C-H stretch | 3074 | 3068 | 3063 | 3066 | 3066 | 3066 | 3253 | 3256–3263 | (3248–3253)[b] |
| | 20 | C-H stretch | 3006 | 3009, 3028 | 3043 | 3032 | 3030 | 3030 | 3188 | 3223–3225 | (3218–3236)[b] |
| | 21 | Ring stretch | 1574 | 1571 | 1571 | 1574 | 1577 | 1577 | 1651 | 1648–1650 | 1644–1654 |
| | 22 | Ring stretch | 1447 | 1446 | 1440 | 1444 | 1443 | 1443 | 1473 | 1482–1483 | 1477–1487 |
| | 23 | C-H wag | 1373 | 1380 | 1384 | 1355 | - | - | 1370 | 1375–1380 | 1377–1389 |
| | 24 | Ring stretch | 1233 | 1230 | 1232 | 1227 | 1227 | 1227 | 1333 | 1334–1336 | 1335–1341 |
| | 25 | C-H wag | 1155 | 1165 | 1155 | 1155 | - | | 1163 | 1169–1170 | 1167–1177 |
| | 26 | C-H wag | 1068[a] | 1065 *, 1072 * | 1064 | 1072 | 1068 | | 1082 | 1079, 1094, 1095, 1097[b], 1098, 1104 | (1167–1177) |
| | 27 | Ring bend | 651 | 649 | - | 649 | 654 | | 668 | 660–663 | 660–667 |

[a] Overlapped with Cl-O modes; [b] A$_1$ and B$_2$ modes are coupled; [c] A$_2$ and B$_1$ modes are coupled.

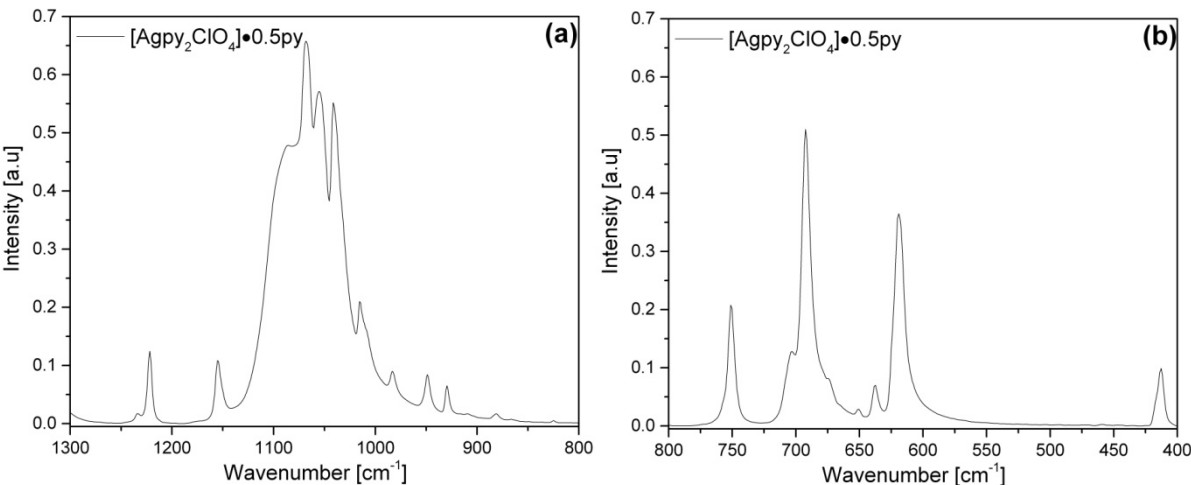

**Figure 9.** IR spectra of compound **1** at room temperature between 1300 and 800 cm$^{-1}$ (**a**) and 800 and 400 cm$^{-1}$ (**b**).

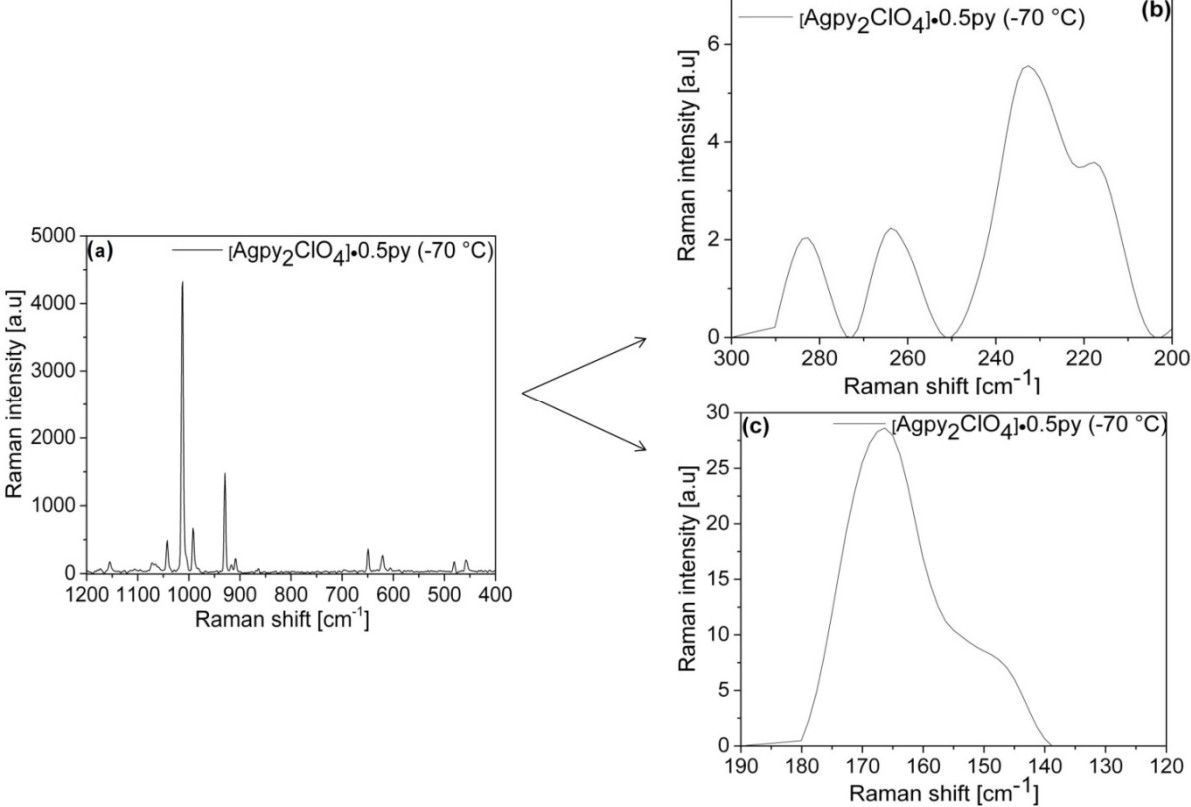

**Figure 10.** Raman spectra of compound **1** at 203 K between 1200 and 400 cm$^{-1}$ (**a**), 300 and 200 cm$^{-1}$ (**b**), and 190 and 130 cm$^{-1}$ (**c**).

The antisymmetric stretching band was shown to be an intensive mode in the IR spectra. A triplet-like band system appeared with maxima at 1041, 1055 and 1069 cm$^{-1}$. This complex band may also include pyridine vibrational modes such as the $\nu_8(A_1)$ ring bending of pyridines; $\nu_7$ (A$_1$) and $\nu_{14}$(B$_2$) C-H wags, both for coordinated and solvate pyridine; and the $\nu_2^{a,b}$(Cl-O) and $\nu_4$(Cl-O) (458 plus 619 cm$^{-1}$ and 474 plus 619 cm$^{-1}$) combination bands at the higher wavenumber side of the complex band system (Figure 9). The coupling with the pyridine C-H wagging was supported by the DFT calculations. For the dimer, the DFT prediction for this mode was a doublet at 1079 (w) and 1088 cm$^{-1}$ (vs), again at larger frequencies than in the experiment. In the former, the antisymmetric vi-

brations of the two perchlorate ions were found to be coupled in-phase with each other and with the $\nu_7$ (A$_1$) wagging mode of pyridine, while in the latter, the perchlorate vibrations were found to be in anti-phase and coupled with the $\nu_{14}$(B$_2$) C-H wag. A large intensity was predicted for the latter mode. For the dimer and 2py unit, in which the solvate pyridine was simulated by py molecules connected to the O$_4$ atoms, the asymmetric perchlorate stretch mode was shown to be involved in a series of strong transitions at 1092, 1112, 1117, 1167 and 1172 cm$^{-1}$. These modes contain large contributions from the two mentioned pyridine in-plane C-H wagging modes, in some cases involving all six pyridine rings. According to the DFT calculations, it cannot be excluded that the triplet-looking structure involves perchlorate stretching modes and that some may be associated with pyridine wags.

In the Raman spectrum, the weak bands of $\nu_{as}$(Cl-O) components were expected and observed (ESI Figure S7). The most intense Raman peak appeared at 1013 cm$^{-1}$, which probably belonged to $\nu_{17}$ (B$_2$) pyridine C-H wag mode, whereas a less intense band at 992 cm$^{-1}$ probably belonged to the same mode of the solvate pyridine (Figure 10a). The irregular shape singlet at 1043 cm$^{-1}$ may have belonged to $\nu_3$(Cl-O). The symmetric bending mode $\delta_s$(Cl-O) was shown to have a doublet at 458 and 474 cm$^{-1}$ in the IR spectra (Figure S6b) and at 456 and 484 cm$^{-1}$ in the Raman spectra (Figure 10a). The corresponding DFT frequencies were 436 and 444 cm$^{-1}$. The antisymmetric bending mode ($\delta_{as}$(F$_2$) (601 and 607 cm$^{-1}$ according to DFT) is a possible component of the band that appeared at 619 cm$^{-1}$ in the IR spectrum (Figure 9b), but this band was shown to coincide with the pyridine $\nu_{10}$ (A$_1$) ring bending mode (629 cm$^{-1}$ according to DFT), which causes a change in the shape of this band. The $\nu_{10}$ mode of coordinated pyridine was not found in the IR spectra of compounds **2** and **3** [10], but this mode was assigned at 619 cm$^{-1}$ in the Raman spectrum of [Agpy$_4$]ReO$_4$ [9].

No splitting due to the triplet nature of $\delta_{as}$(Cl-O) could be observed in the IR spectrum. The $\delta_s{}^{a,b}$(Cl-O) and $\delta_{as}$(Cl-O) combination band components completely coincided with the band system containing $\nu_{as}$(Cl-O) and other modes between 1120 and 1000 cm$^{-1}$ (Figure 9a) (1080,1088,1187,1189 according to DFT). The Raman spectra contained a weak band at 621 cm$^{-1}$ that belonged to $\delta_s$(Cl-O) (Figure 10a). The assignment of the normal modes of the perchlorate ion confirmed the appearance of weak but characteristic perchlorate ion combination and overtone bands [10] in the IR spectrum of compound **1**, e.g., the $\nu_s$ and $\nu_{as}{}^{a,b,c}$ band at 1983, 1994 and 2014 cm$^{-1}$ and the $2\delta_s{}^a$(Cl-O) band (2 × 458 cm$^{-1}$) at 916 cm$^{-1}$. However, the $2\delta_s{}^b$(Cl-O) band (2 × 474 cm$^{-1}$) completely coincided with the $\nu_{15}$ band of the pyridine rings (Figure S6d). Very weak overtones of $2\delta_s$ at 917 cm$^{-1}$ and 968 cm$^{-1}$ were observed in the Raman spectrum of compound **1** (Figure 10a). The combination bands of $\delta_s{}^{a,b}$(Cl-O) and $\delta_{as}$(Cl-O) also appeared at 1070 and 1107 cm$^{-1}$, although the band at 1070 cm$^{-1}$ may have contained the $\nu_7$ and $\nu_{26}$ pyridine CH wag modes as well. The $\nu_{as}$(Cl-O) and $\nu_s$(Cl-O) combination bands were not found in the Raman spectrum.

The detailed assignment of pyridine vibrational modes in the IR and Raman spectra of compound **1** is summarized in Table 3, and these modes were also compared with the assignments for compound **4** and free pyridine. The frequencies obtained with DFT were generally close to the experimental results except for the high-frequency C-H stretches and the ring breathing, for which the calculated frequency was 1002 cm$^{-1}$ for the pyridine molecule instead of the experimental 858 cm$^{-1}$; interestingly, however, the approximately 20 cm$^{-1}$ shift in the complex was well-reproduced.

The other normal modes belonging to the cation, especially the Ag-N modes, were mainly found in the far-IR region, so only the Raman data could be evaluated.

The band in the IR spectrum of compound **1** at 412 cm$^{-1}$ is characteristic of the metal-coordinated pyridine ring [10,13,29]. The Raman spectrum was found to contain the $\nu_s$ (Ag-N) and $\nu_{as}$(Ag-N) band of the Agpy$_2{}^+$ cation at 166 and 231 cm$^{-1}$, and the contributions of the Ag-O modes appeared as a shoulder at 150 cm$^{-1}$ and a badly resolved peak at 218 cm$^{-1}$ [29]. The band at 107 cm$^{-1}$ was previously assigned as the $\nu_s$ (Ag-N) band for the Agpy$_4{}^+$ cation and as the band belonging to the Ag-O contribution of the perchlorate ion

in the Raman and IR spectra of [Agpy₄]ReO₄ [9] and compound **4** (4(Agpy₂ClO₄·[Agpy₄]ClO₄) [11,29]. [Agpy₄]ReO₄ was not shown to contain coordinated (perchlorate or perrhenate) anions, and compound **1** was not shown to contain the Agpy₄⁺ cation. Thus, the band at 107 cm⁻¹ in the Raman spectrum (Figure S7b) of compound **1** could only be assigned as the perchlorate contribution via Ag-O coordination, in accord with Bowmaker's results regarding compound **4** [29].

### 3. Materials and Methods

Chemical-grade sodium perchlorate, silver nitrate and pyridine (Deuton-X Ltd., Érd, Hungary) were used in the synthesis experiments following the methods given in [13]. Compound **1** was identified by its powder X-ray diffractogram and chemical analysis (Ag, pyridine and perchlorate content) according to the methods given in [13,30].

All X-ray powder diffraction measurements were conducted on a Philips PW-1050 Bragg-Brentano parafocusing goniometer (Cu anode; 40 kV and 35 mA tube power) supplied with a secondary beam graphite monochromator and a proportional counter. All scans were recorded in the step mode. The XRD diffraction patterns were evaluated with a full profile fitting technique.

The diffraction pattern of the colorless block shape single crystal of compound **1** with the size of 0.50 mm × 0.30 mm × 0.30 mm was measured on a RIGAKU RAXIS Rapid II diffractometer at 103(2) K using MoK$\alpha$ radiation. A crystal of compound **1** was mounted on a loop with parathon oil. Cell parameters were determined by least squares using 53,600 reflections ($3.055° \leq \theta \leq 27.445°$). Intensity data were collected on a RIGAKU RAXIS-RAPID II diffractometer (graphite monochromator; Mo-$K\alpha$ radiation, $\lambda$ = 0.71073 Å) at 103(2) K in the range $3.141 \leq \theta \leq 27.463$ [25]. A total of 58650 reflections were collected, of which 3347 were unique ($R$(int) = 0.0305, $R(\sigma)$ = 0.0117); intensities of 3257 reflections were greater than $2\sigma(I)$. Completeness to $\theta$ was 0.998. Multi-scan absorption correction was applied to the data (the minimum and maximum transmission factors were 0.717 and 1.000).

The structure was solved by direct methods [31]. Anisotropic full-matrix least-squares refinement [32] on $F^2$ for all non-hydrogen atoms yielded $R_1$ = 0.0253 and $wR_2$ = 0.0600 for 1332 reflections [$I > 2\sigma(I)$] and $R_1$ = 0.0265 and $wR_2$ = 0.0606 for all (3347) intensity data (number of parameters = 191; goodness-of-fit = 1.205; and the maximum and mean shift/esd were 0.001 and 0.000, respectively). The maximum and minimum residual electron density in the final difference map were 0.916 and −0.623 e.Å⁻³, respectively. The weighting scheme applied was $w$ = 1/[$\sigma^2(F_o^2)$ + (0.02228.5193P)² + 8.5193P] where $P = (F_o^2 + 2F_c^2)/3$.

Hydrogen atomic positions were calculated from assumed geometries. Hydrogen atoms were included in structure factor calculations, but they were not refined. The isotropic displacement parameters of the hydrogen atoms were approximated from the $U$(eq) value of the atom they were bonded to.

Crystal data and details of structure determination and refinement are listed in Tables 1 and S1. Atomic coordinates and equivalent isotropic displacement parameters, hydrogen coordinates and equivalent isotropic displacement parameters, anisotropic displacement parameters, bond lengths and angles, and intermolecular interactions of compound **1** are presented in Tables 2 and S1–S6. CCDC-2191559 (compound **1**) contains the supplementary crystallographic data for this paper. These data can be obtained free of charge from The Cambridge Crystallographic Data Centre via www.ccdc.cam.ac.uk/data_request/cif (accessed on 19 July 2022).

The FT-IR spectrum of compound **1** was recorded on the Jasco FT-IR-4600 system described in [32]. The apparatus was equipped with a single reflection diamond ATR accessory (incident angle 45°). The measurements were conducted in the 4000–400 cm⁻¹ region with a resolution of 4 cm⁻¹, and 64 individual spectra were coadded. An ATR correction (OPUS v6.0, Bruker Optik Gmbh, Rudolf-Plank-Str. 27, 76275 Ettlingen, Germany) was performed on the raw spectra.

The Raman measurements of freshly prepared compound **1** were performed on a Horiba Jobin/Yvon LabRAM microspectrometer with an external Nd-laser source (532 nm, ~40 mW, Olympus BX-40 optical microscope) at 203 and 243 K with a Linkam THMS600 temperature-controlled microscope stage). The laser beam was focused (objective of 50×), and a D2 intensity filter was used to prevent thermal decomposition due to laser power. The confocal hole of 1000 μm and a 1800 groove mm$^{-1}$ grating monochromator were used in a confocal system and for light dispersion. The spectra were measured between 4000 and 100 cm$^{-1}$ with a resolution of 3 cm$^{-1}$. Exposure time was 60 s.

Quantum chemistry calculations were performed with the Gaussian 09 suite of programs [33]. The structure and vibrational spectra of characteristic fragments of the elementary cell were determined using density functional theory. We used the B3LYP [34], CAM-B3LYP [35] and M05 functionals [36]. For systems involving transition metals, M05 is the most appropriate, so we report the results obtained with this functional. The difference of the structural parameters obtained with the three functionals showed the usual, relatively small differences. The LANL2DZ basis set and pseudopotential were used for the Ag atom. For the remaining atoms the 6–31 G(d,p) basis set was employed. The presence of the polarization function proved to be essential for the chlorine atom. It is worth noting that when the LANL2DZ basis set was used for all atoms, the geometry optimization led to a formal Cl-O bond rupture within the perchlorate groups, because of the lack of *d* orbitals in the hypervalent Cl atom.

## 4. Conclusions

Compound **1** ([Agpy$_2$ClO$_4$]·0.5py) was successfully prepared via the trituration of [Agpy$_2$ClO$_4$] and 4[Agpy$_2$ClO$_4$]·[Agpy$_4$]ClO$_4$ (as the source of solvate pyridine) mixture in a mixed solvent of acetone:benzene = 1:1 (v = v) at room temperature. The monoclinic crystals of compound **1** were found to be isomorphic with the analogous permanganate complex. Two [Agpy$_2$]$^+$ cations were shown to form a dimeric unit [Agpy$_2$ClO$_4$]$_2$, and silver ions were found to be connected to two ClO$_4^-$ anions via oxygen atoms. The Ag···Ag distance was calculated as 3.3873(5) Å, whereas the Ag···O distances were calculated as 2.840(2) Å and 2.8749(16) Å in the centrosymmetric rectangle of Ag-O-Ag-O. The stoichiometric ratio of the monomer [Agpy$_2$ClO$_4$] and the solvent pyridine was 1:0.5. The guest pyridine occupied 527.2 Å$^3$, which was 18.0% of the volume of the unit cell. There was no additional residual solvent-accessible void in the crystal lattice. The solvate pyridine was connected via its $\alpha$-CH to one of the O atoms of the perchlorate anion. The IR and Raman bands were assigned for all of the perchlorate and pyridine vibrational modes based on correlation analysis and quantum chemistry calculations.

**Supplementary Materials:** The following supporting information can be downloaded at: https://www.mdpi.com/article/10.3390/inorganics10090123/s1, Figure S1: Comparison of the experimental (a) and calculated (from SXRD data) (b) powder XRDs of compound 1. Table S1: Crystal data and details of the structure determination and refinement of compound 1. Table S2: Atomic coordinates (x 104) and equivalent isotropic displacement parameters (Å2 x 103) of compound 1. *U*(eq) is defined as one third of the trace of the orthogonalized *Uij* tensor. Table S3: Hydrogen coordinates (x 104) and isotropic displacement parameters (Å2 x 103) of compound 1. Table S4: Anisotropic displacement parameters (Å2 x 103) of compound 1. The anisotropic displacement factor exponent takes the form: -2θ2(h2a*2U11 + ... + 2hka*b*U12). Table S5: Bond lengths [Å] in compound 1. Table S6: Bond angles [o] in compound 1. Table S7: Molecular geometry of the Agpy2ClO4 dimer calculated using the M05 functional with LANL2DZ basis set and pseudopotential on Ag and the 6-31G(d,p) basis set on the rest of the atoms. Table S8: Molecular geometry of the Agpy2ClO4 dimer coordinated by two pyridine ligands at the O4 atoms, calculated using the M05 functional with LANL2DZ basis set and pseudopotential on Ag and the 6-31G(d,p) basis set on the rest of the atoms. Figure S2: The packing arrangement of compound 1 (capped sticks) and the solvent pyridine (ball and stick rep-

resentation). a., View from the *a* crystallographic axis. b., View from the *b* crystallographic axis. c., View from the *c* crystallographic axis. Figure S3: External vibrational modes of perchlorate ion in compound 1. Figure S4: External vibrational modes of pyridines at C1 site (coordinated ones) in compound 1. Figure S5: External vibrational modes of pyridine at C2 site (solvate) in compound 1. Figure S6: Correlation diagram for perchlorate ion internal modes in compound 1. Figure S7: The correlation diagrams for the hindered translations of Ag$^+$ cations at *C*1 sites. Figure S8: IR spectrum of compound 1 at room temperature between 4000 and 400 cm$^{-1}$ (a). The enlarged spectrum parts are given between 490 and 430 cm$^{-1}$ (b) and 1650 and 1200 cm$^{-1}$ (c), and 3200 and 1650 cm$^{-1}$ (d). Figure S9: Raman spectra of compound 1 between 4000 and 100 cm-1 at 243 and 203 K (a). The enlarged spectrum parts recorded at 203 K are given between 1700 and 100 cm$^{-1}$ (b), 3300-3000 cm$^{-1}$(c).

**Author Contributions:** Conceptualization, L.K.; formal analysis, V.M.P. and P.B.; investigation, N.V.M., N.B., K.A.B., A.F. and G.L.; writing—original draft preparation, L.K.; writing—review and editing, N.V.M., P.B., V.M.P., K.A.B. and G.L.; visualization, N.V.M. and K.A.B.; supervision, L.K.; All authors have read and agreed to the published version of the manuscript.

**Funding:** This research was funded by Hungarian Scientific Research Found, grant number K-124544 and the New National Excellence Program of the Ministry for Innovation and Technology from the source of the National Research, Development and Innovation Fund The APC, grant number UNKP-21-3.

**Institutional Review Board Statement:** Not applicable

**Informed Consent Statement:** Not applicable

**Data Availability Statement:** Not applicable

**Acknowledgments:** P.B. and N.V.M. are grateful for the Hungarian Scientific Research Found (K-124544); K.A.B. expresses his thanks for the ÚNKP-21-3 New National Excellence Program of the Ministry for Innovation and Technology from the source of the National Research, Development and Innovation Fund.

**Conflicts of Interest:** The authors declare no conflict of interest.

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
