# Peer review of "Structure and Vibrational Spectra of Pyridine Solvated Solid Bis(Pyridine)silver(I) Perchlorate, [Agpy2ClO4]·0.5py"

_inorganics, doi:10.3390/inorganics10090123_

Round 1
Reviewer 1 Report
This is another paper of an obviously very productive group and, as usual, work is extremely thorough and certainly warrants publication. As usual, I think the vibrational spectroscopy part is too elaborate, but that is just my opinion subject to my scientific background.
It is also obvious that the authors have done a plethora of work that is relevant to the present article which, thus, should be referenced. In a bigger picture, it would be helpful to a not so specialized reader to have at least some references to silver coordination chemistry. There has been a special issue of the Australian Journal of Chemistry in 2006 (see attached pdf), and maybe also the book chapter:
G. Meyer, M. Sehabi, I. Pantenburg: Coordinative Flexibility of Monovalent Silver, in Design and Construction of Coordination Polymers (Mao-Chun Hong, Ling Chen, editors), John Wiley and Sons, in [AgI←L1]L2 Complexes 2009, chapter 1, 1-23.
might be of some relevance.

Author Response
First of all, we would like to express our sincere thanks to the reviewers for their suggestions that helped us to increase the quality of our manuscript.
Responses given for remarks/questions are given in italic below.
This is another paper of an obviously very productive group and, as usual, work is extremely thorough and certainly warrants publication. As usual, I think the vibrational spectroscopy part is too elaborate, but that is just my opinion subject to my scientific background.
A part of the vibrational spectroscopic results (some correlation analysis tables) has been moved to ESI.
It is also obvious that the authors have done a plethora of work that is relevant to the present article, which, thus, should be referenced. In a bigger picture, it would be helpful to a not so specialized reader to have at least some references to silver coordination chemistry. There has been a special issue of the Australian Journal of Chemistry in 2006 (see attached pdf), and maybe also the book chapter:
Meyer, M. Sehabi, I. Pantenburg: Coordinative Flexibility of Monovalent Silver, in Design and Construction of Coordination Polymers (Mao-Chun Hong, Ling Chen, editors), John Wiley and Sons, in [AgI←L1]L2 Complexes 2009, chapter 1, 1-23.
might be of some relevance.
Several papers mentioned have been inserted, and the introduction has been modified according to that.
Reviewer 2 Report
In my opinion, this paper essentially represents crystal structure report of a one more complex of [Agpyn]XO4 type. Previously, a number of similar complexes was already described in details. From a structural point of view, the presented complex is unremarkable against the background of other representatives of this family. Significantly, the introduction does not explain in any way why this complex deserves attention and a separate study. The rest of the paper is devoted to the routine assignment of bands in the Raman and IR spectra of the presented complex. Again, the assignments made do not bring scientific novelty to justify publication in this journal. Moreover, the band assignments are made without the use of quantum chemistry methods and thus are not completely convincing.
In summary, I cannot recommend the publication of this work in Inorganics.
Author Response
The answers for remarks are given in italic below:
In my opinion, this paper essentially represents crystal structure report of a one more complex of [Agpyn]XO4 type. Previously, a number of similar complexes was already described in details. From a structural point of view, the presented complex is unremarkable against the background of other representatives of this family.
There are four basic types of [Agpyn]XO4 complexes with n=2, 2.4, 2.5, and 4, which differ in their structure and topology very much. We prepared a missing member of the series with X=Cl, the second member of the sub-group (n=2.5) discovered by us recently. The structural motifs and topology depending on the anion in these complexes and the secondary interactions (argentophilic, hydrogen bond, stacking, etc.) are very interesting fields for crystal engineers, and we studied the effects of these comparing the only existing analog, the permanganate salt (prepared by us). We also compared the structure of the title compound with other three existing pyridinesilver perchlorate.
I found it unacceptable that the reviewer qualifies this structure to be "unremarkable", because it is only the second member of the compound family has this kind of structure. The properties and structural motifs of the title compound and the reason for its formation in the AgClO4-py system where only the other three compounds could be isolated -and we could find a new one - and solving the structure to get information about that - may not be unremarkable.
Significantly, the introduction does not explain in any way why this complex deserves attention and is a separate study.
We gave information about the structural variability of AgpynXO4 compounds and declared that the structure of a missing member of the series was given in the introduction.
The other three known pyridinesilver perchlorate structures were studied earlier by other authors many years ago – so they could not solve the structure of the title compound because they did not know that. When we prepared the title compound, the structures of other three were known. So, we cannot understand how we could write about this compound “not separatedly”. The single crystal study about permanganate analog was studied more than 10 years ago (it was published 4 years ago only), but at that time the single crystals for study the structure of the title compound (due to its challenging preparation and crystal growing) was not available.
The rest of the paper is devoted to the routine assignment of bands in the Raman and IR spectra of the presented complex. Again, the assignments made do not bring scientific novelty to justify publication in this journal.
The band assignment has been done to show the effect of anion-cation and other secondary interactions and studied whether it is possible to distinguish the solvate/coordinated pyridine normal modes or not. We have done a correlation analysis for the compound, showing the different number of bands of the two types of pyridines can be expected.
Furthermore, we compared the spectral characteristics of the compound with other members of the pyridinesilver complexes to solve a questionable assignment of an Ag-N mode of Agpy4 cation and the contribution of the Ag-O modes of the coordinated anion to the Agpy2-cationic unit. We compared the spectral data of Agpy4ReO4 compounds without anion coordination (no Ag-O) contribution and the Raman spectrum of the title compound (it is an Ag-O anion coordination) with the IR and Raman spectra of [4Agpy2ClO4].[Agpy4]ClO4, which contains py2Ag..O-Cl coordination and isolated Agpy4 cations without perchlorate coordination as well.
Moreover, the band assignments are made without the use of quantum chemistry methods and thus are not completely convincing.
We could assign the questionable band without quantum chemical calculations according to the spectroscopic evaluation given above. Since the differences between the solvate and coordinate pyridine modes were found mostly to be too small to distinguish them from each other, using quantum chemical calculations looked like useless (we did ab initio calculations on a similar pyridinesilver permanganate complex, and the error in the calculated band positions were bigger than the expected differences between the band positions of coordinated and non-coordinated pyridines.
The perchlorate modes could be unambiguously identified on the base of changes in intensities of bands belonging to the symmetric/antisymmetric modes and their splittings. In order to do it, we had to do a detailed spectroscopic analysis.
In summary, I cannot recommend the publication of this work in Inorganics.
We feel that on the basis of the Reviewer’s major arguments in his/her comments, the majority of papers in the field of inorganic chemistry could not be suggested for publication. I am afraid that this perception would prevent revealing and spreading important data on many classes of inorganic compounds.
Reviewer 3 Report
Nóra V. May et al., explored the experimental study of [Agpy2ClO4]·0.5py. The manuscript looks good. Minor revision is required on the basis of following comments.
1. In abstract, kindly write the lattice parameters and space group.
2. Kindly reduce keywords to five.
3. Page 2, line 81, kindly write the symmetry code with which half pyridine is formed or generated.
4. In Figure 1, kindly correct the label of symmetry related atoms C32 and C33 as C32i and C33i. In caption, kindly specify symmetry code.
5. Quality of Figure 5,6, 13 and 14 is not good. Kindly redraw them.
6. Kindly reduce number of figures to 10. Move some Figures to supplementary information file.
7. In introduction section, kindly specify the importance and the applications of such class of compounds. For example, see the following crystal structures containing pyridine ring. https://doi.org/10.3390/cryst10090778
Author Response
First of all, we would like to express our sincere thanks to the reviewers for their suggestions that helped us to increase the quality of our manuscript.
Nóra V. May et al., explored the experimental study of [Agpy2ClO4]·0.5py. The manuscript looks good. Minor revision is required on the basis of following comments.
- In abstract, kindly write the lattice parameters and space group.
It has been done.
- Kindly reduce keywords to five.
It has been done.
- Page 2, line 81, kindly write the symmetry code with which half pyridine is formed or generated.
It has been done.
- In Figure 1, kindly correct the label of symmetry related atoms C32 and C33 as C32i and C33i. In caption, kindly specify symmetry code.
It has been done.
- Quality of Figure 5,6, 13 and 14 is not good. Kindly redraw them.
It has been done.
- Kindly reduce number of figures to 10. Move some Figures to supplementary information file.
It has been done.
- In introduction section, kindly specify the importance and the applications of such class of compounds. For example, see the following crystal structures containing pyridine ring. https://doi.org/10.3390/cryst10090778
It has been done.
Round 2
Reviewer 2 Report
In revision, the authors did not respond to the reviewer's comments. I still think that this paper is more suitable for a journal devoted to crystallography and structural chemistry, rather than Inorganics.
I still have about the same criticisms to this paper:
1. The introduction is too laconic and does not answer the two key questions: why are these compounds interesting? What interesting properties do they have?
2. Again, very similar analogues of the presented complex previously were already studied, including isostructural permanganate-based complex. Thus, I see no reason for a detailed analysis for another similar complex.
3. Now, DFT calculations are the gold standard for accurate band assignment in vibrational spectra. Such calculations have long been routine for small molecules. The phenomenological approach used by the authors is outdated and does not provide high accuracy in assigning vibrational spectra. Therefore, authors should perform the DFT calculations before submitting paper in other journal.
4. The authors did not investigate any functional properties for the presented complex. Meanwhile, almost all articles in this journal include a study of the properties of synthesized compounds, e.g. luminescence, biological activity, conductivity, electrochemistry, etc.
To sum up, I can't recommend the publication of this paper in Inorganics.
Author Response
First, I would like to reflect to the first sentence of the Reviewer’s report:
“In revision, the authors did not respond to the Reviewer's comments” As I quote it below, we did give answers to the comments of the Reviewer. . We are afraid he/she wanted to hear something else, probably more formal issues than the scientific argument we provided. The Reviewer repeated his/her criticism and insist of his/her opinion. I feel that the Reviewer has a different attitude from ours and could not change his/her mind, independently of the content of our answers. We think that what we do is of academic interest while the Reviewer does not appreciate the kind of qualitative information such as what factors drive the formation of crystals. To shows that we did reply to the comment, we highlighted in yellow the Reviewer’s comments in the first report, and our answers in green. I have to note that already the first sentence of the comment below is not really correct: only one complex is known which has a similar structure – and not “a number of complexes."
In my opinion, this paper essentially represents crystal structure report of a one more complex of [Agpyn]XO4 type. Previously, a number of similar complexes was already described in details. From a structural point of view, the presented complex is unremarkable against the background of other representatives of this family.
There are four basic types of [Agpyn]XO4 complexes with n=2,2.4, 2.5, and 4, which differ in their structure and topology very much. We prepared a missing member of the series with X=Cl, the second member of the sub-group (n=2.5) discovered by us recently. The structural motifs and topology depending on the anion in these complexes and the secondary interactions (argentophilic, hydrogen bond, stacking, etc.) are very interesting fields for crystal engineers, and we studied the effects of these comparing the only existing analog, the permanganate salt (prepared by us). We also compared the structure of title compound with other three existing - but very different in their composition and structure - pyridinesilver perchlorate.
I found it unacceptable that the Reviewer qualifies this structure to be "unremarkable", because it is only the second member of the compounds to have this kind of structure. The properties and structural motifs of the title compound and the reason for its formation in the AgClO4-py system where only the other three compounds could be isolated -and we could find a new one - and solving the structure to get information about that - may not be unremarkable.
Significantly, the introduction does not explain in any way why this complex deserves attention and is a separate study.
We gave information about the structural variability of AgpynXO4 compounds and declared that the structure of a missing member of the series was given in the introduction.
The other three known pyridinesilver perchlorate structures were studied earlier by other authors many years ago – so they could not solve the structure of the title compound because they did not know that. When we prepared the title compound, the structures of other three were known. So, we cannot understand how we could write about this compound “not separatedly”. The single crystal study about permanganate analog was studied ca. 15 years ago (it was published 4 years ago only), but at that time the single crystals for study the structure of the title compound (due to its challenging preparation and crystal growing) was not available.
The rest of the paper is devoted to the routine assignment of bands in the Raman and IR spectra of the presented complex. Again, the assignments made do not bring scientific novelty to justify publication in this journal.
The band assignment has been done to show the effect of anion-cation and other secondary interactions and studied whether it is possible to distinguish the solvate/coordinated pyridine normal modes or not. We have done a correlation analysis for the compound, showing the different number of bands of the two types of pyridines can be expected.
Furthermore, we compared the spectral characteristics of the compound with other members of the pyridinesilver complexes to solve a questionable assignment of an Ag-N mode of Agpy4 cation and the contribution of the Ag-O modes of the coordinated anion to the Agpy2-cationic unit. We compared the spectral data of Agpy4ReO4 compounds without anion coordination (no Ag-O) contribution and the Raman spectrum of the title compound (it is an Ag-O anion coordination) with the IR and Raman spectra of [4Agpy2ClO4].[Agpy4]ClO4, which contains py2Ag..O-Cl coordination and isolated Agpy4 cations without perchlorate coordination as well.
Moreover, the band assignments are made without the use of quantum chemistry methods and thus are not completely convincing.
We could assign the questionable band without quantum chemical calculations according to the spectroscopic evaluation given above. Since the differences between the solvate and coordinate pyridine modes were found mostly to be too small to distinguish them from each other, using quantum chemical calculations looked like useless (we did ab initio calculations on a similar pyridinesilver permanganate complex, and the error in the calculated band positions were bigger than the expected differences between the band positions of coordinated and non-coordinated pyridines.
The perchlorate modes could be unambiguously identified on the base of changes in intensities of bands belonging to the symmetric/antisymmetric modes and their splittings. In order to do it, we had to do a detailed spectroscopic analysis.
In summary, I cannot recommend the publication of this work in Inorganics.
We feel that on the basis of the Reviewer’s major arguments in his/her comments, the majority of papers in the field of inorganic chemistry could not be suggested for publication. I am afraid that this perception would prevent revealing and spreading important data on many classes of inorganic compounds.
Answers to the present comments are below and marked in italics.
Comment:
I still think that this paper is more suitable for a journal devoted to crystallography and structural chemistry, rather than Inorganics.
We think that the correlation analysis, IR and low-temperature Raman measurements and the assignment of vibrational spectra do not really belong to the field of crystallography – they use the determined structure but go far beyond structure determination.
The scope of “Inorganics” is copied from the website (we highlighted with green the parts to which our paper belongs): No limitation is given, which would say that all the fields should be present in one paper.
Topics include but are not limited to:
- synthesis and characterization of inorganic compounds, complexes and materials
- structure and bonding in inorganic molecular and solid state compounds
- spectroscopic, magnetic, physical and chemical properties of inorganic compounds
- chemical reactivity, physical properties and applications of inorganic compounds and materials
- mechanisms of inorganic reactions
- organometallic compounds
- inorganic cluster chemistry
- heterogenous and homogeneous catalytic reactions promoted by inorganic compounds
- thermodynamics and kinetics of significant new and known inorganic compounds
- supramolecular systems and coordination polymers
- bio-inorganic chemistry and applications of inorganic compounds in biological systems and medicine
- environmental and sustainable energy applications of inorganic compounds and materials
Comment:
I still have about the same criticisms to this paper:
- The introduction is too laconic and does not answer the two key questions: why are these compounds interesting? What interesting properties do they have?
We did modify the Introduction to guide the reader to realize that the compound we report on does have some interesting properties. We note that the compound is new and was very hard to synthesize and in such a case everything is new and interesting information about that.
To know the structure and basic spectroscopic properties of a new compound is an essential thing, with giving the exact atom positions and interactions between the cation, anion, and solvate parts. Previously we gave powder XRD only , which showed isomorphism, but it did not give detailed properties and exact data about the structure (e.g., about hydrogen bonds and other interactions found). It is not clear for us why the Reviewer talk sabout known "analogs", because only one (1 !) analog compound exists – as we stated twice until now, and do it the third time now. The structure of that compound - Agpy2MnO4.05.py has been determined, but the spectroscopy was not given in detail. The correlation analysis based on the newly determined crystallographic data was not preformed for the previously known analog, and it is novel and available only for the perchlorate reported in this manuscript.
We gave a detailed explanation that there were three known compounds in the AgClO4-py system, and we prepared the fourth one. A new one. The interactions between the perchlorate anion and cation/solvate were determined and compared with the interactions of permanganate in the permanganate compound. Without determining the structure, it couldn't have been possible to compare and analyze the effect of the anions on these interactions and the cation conformation.
Comment:
Again, very similar analogues of the presented complex previously were already studied, including isostructural permanganate-based complex. Thus, I see no reason for a detailed analysis for another similar complex.
Again, it is not clear for us how many “analogs" are referred to, because we are aware only one. The compound we study is a solvate with a curious structure. In addition, for the previously known analog, the permanganate, spectroscopy was not studied in detail not to mention correlation analysis. In fact, the only compound with species in the site's characteristics only to this (and to the single analog, the permanganate compound) for which this kind of analysis has been performed is the solvate we report in this paper.
If the Reviewer was right, since sodium chloride structure was determined a long time ago, practically as the first among the inorganic compounds, later, should not have studied the properties of the sodium bromide or sodium iodide? Since the potassium aluminum alum structure and properties had been known, any other alum structures (many dozens) and properties to compare and determine the effect of cations in should not have been studied.
The Reviewer seems not to have appreciated this, which we think shows that his approach is not very professional and incorrect. As this criticism is repeated in the second report, disregarding our attempt to make it clear in our answer to the first criticism , we are afraid the Reviewer has made his mind earlier and does not wish to change it..
Comment:
Now, DFT calculations are the gold standard for accurate band assignment in vibrational spectra. Such calculations have long been routine for small molecules. The phenomenological approach used by the authors is outdated and does not provide high accuracy in assigning vibrational spectra. Therefore, authors should perform the DFT calculations before submitting paper in other journal.
It seems to us that the Reviewer disregards that vibrational analysis and assignments for bands were successfully done even before using quantum chemistry based on chemical group theory and spectroscopical analyses. As we explained in detail, the assignment of the bands could be done with spectroscopic methods with complete reliability. In fact, this kind of analysis provides more qualitative information than the purely numerical quantum chemical calculations., this contributing more to _understanding_ the properties.
To convince the Reviewer, we did density functional theory calculations for our compound. The results confirm the assignments performed without quantum chemistry. The results also reflect that quantum chemistry can _help_ the assignment of spectra of a solid/state structure, but the calculated frequencies never match the experimental ones throughout the whole spectrum, and quantum chemist also rely on common sense and rules of thumb.
Comment:
- The authors did not investigate any functional properties for the presented complex. Meanwhile, almost all articles in this journal include a study of the properties of synthesized compounds, e.g. luminescence, biological activity, conductivity, electrochemistry, etc.
We are afraid the Reviewer follows the standard reviewing pattern and does not pay attention to the peculiarity of the compound that is the subject of this study. In particular, our compound is a solvate. It loses its extra pyridine easily on standing or on heating. We have not studied its thermal properties because the first step is the loss of the solvate pyridine, and then the decomposition characterizes Agpy2ClO4 (which we published earlier).
Similarly, if the title compound would be dissolved to study the electrochemical or biological properties, it would decompose immediately into Agpy2- perchlorate, and pyridine, and we studied the properties of this and not the title compound.,The novelty this compound provides is not some practical applicability (it is not stable for that), but that its structure shows that not only Agpy4 cations but also pyridine solvent molecules can be positioned between the Agpy2+ and perchlorate ion layers to make structures stabilized by different secondary interactions between the same components.
Comment:
To sum up, I can't recommend the publication of this paper in Inorganics.
We modified the paper according to the criticism of the Reviewer. We even performed quantum chemistry calculations that, as we expected, confirms the empirical spectrum assignment. We argued above that we think the paper matches more the topics covered by Inorganics than those of a crystallographic journal. We hope that when the Editor reviews the report of the Reviewer and our replies, will agree with us. We trust her wisdom when making the decision about the acceptability of our paper.